# Assessing the impact of environmental laws and technological advancements on carbon dioxide mitigation in China's mining industry

Dongxiao Xu[1‡], Weijun Chen 🄳[1‡*], Yunzhe Chen[2], Jing Wen[3]

1 South China Agriculture University, School of Humanities and Law, Guangzhou, China, 2 Nanjing University of Science and Technology, European and Asian Studies Institution, Nanjing, China, 3 Tianjin University, Tianjin, China

‡ These authors are co-first authors on this work.
* chenweijun@scau.edu.cn

## Abstract

The present research examines the relationship between China's mineral extraction-related carbon dioxide ($CO_2$) emissions and factors such as the legislative law, openness, green economic growth, FDI, technology innovation, and green patent from 1989 to 2020. Depending on statistics from the China Statistical Yearbook as well as other global databases, it finds the legislative law and openness contribute to sustainable mineral extracting in China. The efficacies of legislative laws can demonstrate by the substantial decreases in $CO_2$ that occur in response to a one percent rise in these variables. Reducing carbon dioxide emissions is another positive association with the green growth index. The absence of green patents, innovation (patent applications), and foreign direct investment (FDI) unexpectedly reveals consequences on the environment. Improving the long-term sustainability of China's mineral extracting sector should be a priority for policymakers. To achieve this, we need to reinforce legal frameworks, encourage green economic growth (GEG), integrate foreign direct investment (FDI) with sustainable methods, incentivised green innovations and promote the import of green technology.

## 1. Introduction

Two crucial areas have received a lot of attention from studies, politicians, and executives: making the most of Earth's resources and implementing a marketing plan in the hard financial environment of the present. Better management of the environment's assets can increase product availability, decrease global harm risks, increase user contentment, and maintain compliance with regulations [1]. On the other hand, entrepreneurial plans are what really make a difference when it comes to driving innovation, finding business opportunities, and deciding how well or poorly an organisation does in the end. Companies can provide solutions to the complex

**Data availability statement:** All relevant data are within the paper.

**Funding:** The author(s) received no specific funding for this work.

**Competing interests:** The authors have declared that no competing interests exist.

problems of mineral extracting, environmental protection, economic development, and green technology; hence, the entrepreneurial approach considered [2]. Their creative thinking and entrepreneurial endeavors may pave the way for more equitable and cost-effective management of biodiversity. "Organic asset maintenance" refers to efforts made to make appropriate and suitable use of an area's biological resources in order to extend their useful life. Most businesses require the careful handling of organic assets because they are fundamental to their functioning. Natural resource management is essential to producing ecological outcomes for many reasons, the most important of which is that it facilitates the ethical and environmentally responsible operation of businesses [3]. Using natural resources responsibly allows organisations to fulfill their ethical obligation to operate while lowering their economic impact and risks. The environmental and carbon footprints of organisations reduced through sustainability initiatives like water conservation, green power usage, and waste reduction [4].

Extracting mineral resources, which are important for many different types of industries and the economy as a whole, has recently become one of the most significant drivers of environmental degradation. The mineral extracting sectors are responsible for more than 80% of the world's biodiversity loss and 50% of its carbon emissions, by [5]. According to [6], rainforest and habitat destruction are common outcomes of extracting processes that involve large-scale land clearance. Ecosystems and wildlife are particularly vulnerable to the dangers presented towards them by the heavy metals and toxic substances that released into the air, water, and soil result of mining operations. According to several studies [6], mineral extracting often requires resource-intensive methods, which add to greenhouse gas emissions and deepen concerns about climate change. The effects of unused mines on ecosystems can be long lasting and devastating to landscapes. Further endangering local ecosystems is the practice of disposing of mining waste, which includes tailings, in a way that causes water contamination [7]. Stress the need to promote sustainable mining techniques and reduce the environmental effects in order to meet mineral requirements while also protecting the planet.

Countries must shift towards environmentally friendly procedures in the mining industry because of the extensive contamination and environmental damage that occurs during mineral resource extracting. According to [8], these changes necessitate clear legal frameworks controlling mining operations and strong economic laws to prioritise and enforce environmental protections. One way to lessen mining's impact on the environment is to take advantage of new technology that makes extracting easier and safer. According to [9], encouraging the global implementation of eco-friendly methods and technologies can be achieved by facilitating global trade in sustainable materials used in mineral extracting. Finding a balance between mineral needs and environmental preservation is possible when countries prioritise economic interests alongside ecological responsibility, according to [10]. Protecting the environment and developing a more robust and accountable global economy are both advanced when companies make sustainability a top priority when extracting mineral resources.

According to several studies [11,12], the idea of green economic growth is crucial for promoting sustainable development in the mineral extracting industry. It serves as a driving force to enhance environmentally friendly programs in both the financial and social spheres. The mining sector can play a role in promoting sustainable financial development by adopting environmentally conscious practices. These practices should prioritise sustainable mineral extracting, minimise carbon footprints, and maximise the utilisation of resources [13]. Mineral extracting can have positive effects on society and the environment, in addition to financial ones, if governments and industry players prioritise sustainable financial development. An agenda for sustainable growth must include investing in environmentally friendly mining actions, sustainable energy sources, and green technology. In addition to protecting cultural and environmental artifacts, sustainable economic growth encourages social inclusion by making sure that community in the area gain from mining operations. According to [14], the mineral extracting industry can play a role in bringing about a balance between economic growth and the conservation of the environment by integrating green principles into its operations.

This paper aims to examine the relationship between green economic growth, openness, and the legislative law as it pertains to carbon dioxide ($CO_2$) emissions in China's minerals extracting industry. As the biggest user and manufacturer of minerals on a global scale, China plays a crucial role in both financial and environmental dynamics. China has used 10.8 billion tonnes of fuel, 2.58 trillion cubic meters of gasoline, 83.8 billion tonnes of coal, 13.8 billion tonnes of natural steel, 150 million tonnes of perfected copper, and 342 million tonnes of primary aluminum over the last fifty years [15]. Its rapid economic development has greatly exacerbated the excessive demand for and harmful effects of mineral extracting on the environment. Researching China is essential because the country's socioeconomic climate is complex and has a major impact on international mineral markets. The highly industrialized economic sector of China highlights the need to look into the ecological effects of mineral extracting operations. Communities in close proximity to mining sites in China are dealing with environmental and health concerns at the same time, which is transforming the country's social landscape. It is crucial to comprehend the environmental impact of China's mineral extracting industry due to the country's significant global contributions to waste and carbon dioxide ($CO_2$) emissions. The study takes on more significance and urgency in light of China's sustainability initiatives, such as its carbon balance plan for 2060 and its zero-carbon target for 2030. It is very important to look at how economic growth, the legislative law, and openness affect carbon dioxide emissions from mineral extracting in China in order to make policies that support the country's goals for a sustainable future.

By focusing on the evaluation of the effects of green economic growth, the legislative law, and openness in relation to carbon dioxide ($CO_2$) emissions associated with China's mineral extracting industry, this study significantly adds to the current body of knowledge. Even though there have been a lot of studies on environmentally friendly practices and economic growth, this study is the first to examine the complex interplay between these three critical variables and how they affect carbon dioxide emissions. By examining the mineral extracting industry in China, a major source of emissions worldwide, this study aims to provide a more complex understanding of the ways in which green economic policies, regulatory frameworks, and open government practices influence and mitigate environmental consequences. This paper's framework flows instinctively as it explores deeply the complex interplay among China's mineral extracting sector, pollution, green economic growth, openness, and the legislative law.

The preceding sections are provided as: Section 2, which conducts a thorough literature review. Section 3 conducts an in-depth analysis of the crucial link between environmental degradation and China's mineral extracting industry. Section 4 then provides results and discussion. Section 5 concludes the paper by integrating the main points and findings to provide a thoughtful and conclusive conclusion, policy implication and future study directions.

. According to recent research by Climate Analytics and the International Energy Agency (IEA), world emissions from fossil fuels may peak in 2023 (Fig 1). This is because while the use of fossil fuels is declining, the usage of clean energy is increasing rapidly.

 

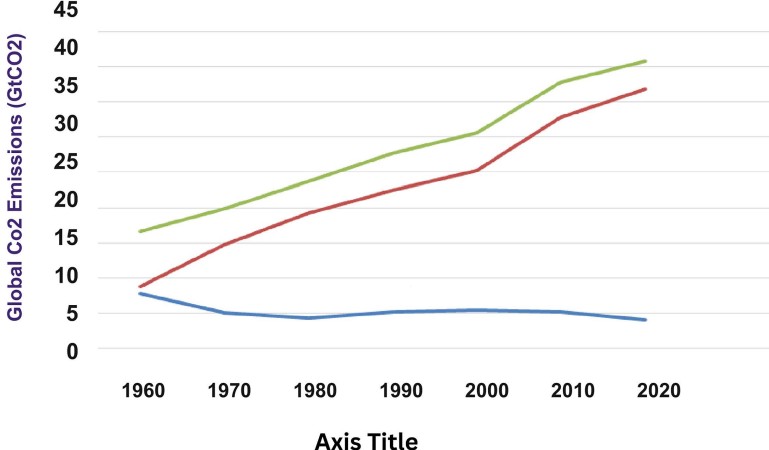

**Fig 1. Global CO2 emission estimate near here.**

## 2. Literature review

Researchers and politicians have long expressed concerns about a certain depletion of natural resources [16]. Deserve special mention for their significant contributions in this area. Several scholars [17], have agreed with them. All natural resources, including Earth's mineral wealth, are limited, according to these so-called resource optimists. They argue that the scarcity of resources will eventually become an issue if demand keeps on rising. Contrarily, some writers [18] are more positive and suggest that technological progress can greatly ease scarcity issues. The fact that mineral resource preserves, as stated by the US Geological Survey, have stayed stable or perhaps increased and that actual mineral resource costs have hardly moved over time lends credence to these resource-optimistic individuals' claims [19]. Even though resource optimists acknowledge that mining for natural resources is going to become more expensive over time, they remain optimistic because they believe in the innovation of the human race to find solutions to such challenges.

In accordance with the Sustainable Development Goals (SDGs), the Paris Agreement, and social obligation, the United Nations Resource Management System (UNRMS) provides a framework for managing and completing resource mining and utilisation operations in accordance with a set of principles and requirements derived from the United Nations Framework Convention on Climate Change (UNFC). During the discussions, the UNFC, this established at UNECE that brought up as a universal language and framework for the classification, management, and reporting of all kinds of energy, mineral, and mineral-based resources, including CRM. Alternative energy sources such as wind, solar, geothermal, and biofuel can all be incorporated into this framework, along with all raw material initiatives, whether they involve primary (like mining) or secondary (like recycling materials, which contribute to the circular economy) raw materials. This enables comparisons made across distinct nations and resources [20].

More and more works have emerged in the years following COVID-19, focusing on the concepts of green economic growth, which seeks to integrate economic revitalization with environmental sustainability recognized by researchers like [21,22] a more resilient and well-rounded strategy for recovery is required. This strategy needs to address both the current environmental threats and the long-term financial effects of the pandemic. According to this literature, moving towards green and low-carbon energy sources is crucial in reducing emissions of greenhouse gases. Creating jobs and improving sustainability in the long run are two goals of green stimulus packages, according to study findings by [23]. Such plans should emphasize investing in clean energy and solutions derived from nature. In addition, research such as that of [24] highlights the significance of incorporating environmentally friendly recovery strategies into financial policies and activities. Green economic growth has been an ongoing conversation since the COVID-19 era began, with many individuals emphasizing the need to rebuild in a sustainable and environmentally friendly way for the betterment of future generations.

The effects of mineral extraction on the natural environment extensively and carefully examined in the literature [25]. Several studies have investigated the effects of mining on the environment, the quality of water, air pollution, ecosystems, and other environmental factors, such as [26]. For instance, conducted a comprehensive examination of the environmental impacts of mining, covering topics like water consumption, greenhouse gas emissions, and energy efficiency [27]. Research such as that of [28] has provided ample evidence of the negative impacts of mining on habitat loss and land degradation extracting.

Second, studies on ethical management and sustainability provide a key context for understanding the function of effective laws and regulations in the extractive minerals business. In particular [29], examine how democratic and accountable leadership improves sustainable behaviors& environmental performance [30]. Emphasize the relevance of regulatory efficacy in reducing resource extracting industry environmental degradation [31]. Also shed light on the connections between legislative systems, laws pertaining to the environment & equitable development, providing a broader perspective of governance structures with environmental results.

Past research looks into the complexities of green economic growth in China, offering useful data about how economic growth and environmental sustainability coincide [32]. Examine China's initiatives to promote green growth in detail, drawing attention to the country's resolve to balance economic development with environmental protection. Their research shows that economic policies must consider environmental factors to achieve equitable and sustainable growth. Building on this idea [33], investigate how green finance helps promote sustainable business practices in China, highlighting the various funding mechanisms that back sustainable projects. In addition [34], provide valuable insights by studying how green innovation affects economic growth; this shows how technological progress can promote sustainable development and economic prosperity.

While academic work on the environmental impacts of mining and sustainable practices is abundant, more research needed that specifically examines the effects of green economic growth, openness, and the legislative law on CO2 emissions in China's mineral extracting industry. Although some research has looked at environmental sustainability and governing structures in isolation, the interplay between these two critical factors, as they pertain to carbon dioxide emissions from mineral extracting, has yet to receive much attention [35]. This study primarily aims to support the hypothesis that China's mineral extracting sector could significantly decrease its carbon dioxide (CO2) emissions through the implementation of strong green economic growth projects, a more robust legislative law, and more transparent governance procedures. Extracting of minerals operations expected to have less of an effect on the environment if robust environmental regulations, transparent legal structures, and inclusive governance methods work together to promote sustainable practices.

## 3. Research methodology

### 3.1. China's mining industry and environmental contamination

A major contributor to China's economic development, the mineral extracting sector has also brought considerable environmental issues, most notably pollution. Consider that in 2020, 4.9% of China's total emissions came from the country's non-metallic industry, which was responsible for the release of about 665 million tonnes of carbon dioxide. Increased mining has contributed to pollution of the environment, water, and soil as a result of fast industrialisation and the unprecedented need for material resources [36]. The contaminants in the air are a direct result of mineral extracting procedures, which include activities like coal mining and metal processing. Soil contamination from mining operations and the release of untreated sewage into rivers both contribute to the deterioration of environmental damage. Because of the highly energy-intensive extracting techniques, carbon dioxide (CO2) emissions contribute significantly to climate change, which has far-reaching consequences for ecosystems around the world. Green mining is a concept that has gained currency across China in the last decade. The goal of various green mining projects is to achieve the best possible financial and environmental results. Following the four stages outlined in the "National Program of Mineral Resources: Developing

China Minerals Green Mine Development Program" China plans to build 661 environmentally friendly mines. To transform the mineral extraction industry into one that is more environmentally sustainable, green economic expansion, effective legislative procedures including the supremacy of law & responsibility, green FDI, innovation assistance, and ecologically friendly methods must integrated into an extensive plan. The unwavering quest for economic growth is a conscious effort to balance the need for ecological protection with the demands of economic progress. It emphasizes a diligent dedication to a development paradigm that acknowledges the inherent interdependence between human activity and environmental health in addition to seeking economic growth. This dedication represents a break from traditional growth patterns, which frequently come at the price of natural ecosystems, and serves as the cornerstone for developing and applying environmentally responsible policies. Enforcing and maintaining appropriate environmental standards in the extracting minerals industry requires transparent governance systems that support the legislative law [37]. Green FDI directs capital towards environmentally beneficial projects, promoting industry sustainability. Supporting patenting and innovation programs simultaneously encourages the creation and application of greener technology, which helps to lessen environmental footprints. The mining sector is moving away from traditional, more polluting technology and towards cleaner energy sources and automation by importing green components, as demonstrated by using renewable energy and electrically powered mining equipment. Over 40% of emissions created by humans have stayed in the atmosphere since the industrial revolution, with the remaining 40% being absorbed by land and ocean sinks. This graphic illustrates how human activity has a cumulatively increasing effect on atmospheric CO2 levels. The percentage of these total emissions that remain in the atmosphere displayed in the lower chart, which adds further context. This emphasizes how, in spite of natural sinks' absorptive capacities, a substantial and ongoing emissions contribute to climate change.

### 3.2. Representation of the data and the model's specifications

Carbon dioxide (CO2) emissions from China's mineral extracting activities are the focus of the present research, which aims to measure the values that represent the consequences of the legislative law, openness, and green economic growth. This data derived from the China Statistical Yearbook and other worldwide databases covering the period from 1989 to 2020. In this study, the mineral extracting industry's carbon dioxide emissions will serve as the dependent factor. According to [38], the explanatory factors contain measures for the legislative law, openness, and an inclusive green growth index. In particular, this index looks at how China is doing in terms of economic growth, preserving the environment, and social equity. Our empirical framework includes control factors like FDI, innovation (patent applications), and the acquisition of sustainable technologies for mineral extracting in line with the transmission mechanisms described in the previous section. Table 1 provides a comprehensive summary of the empirical approach to selected variables extracting.

The predicted patterns of the chosen factors regarding carbon emissions because of the extracting of minerals based on how each affects environmental results. The Legislative law and Openness components predicted to trend positively, indicating that lower CO2 emissions will probably result from stronger regulatory structures coupled with open governance. Openness encourages responsibility and environmentally conscious behavior while obeying the regulations of laws could improve compliance with regulatory laws. Lower CO2 emissions are associated to greater ratings on the Green Growth Index (GGI), which measures social equality, the preservation of the environment, & financial growth. Foreign direct investment expected to boost mining for minerals using environmentally sustainable technologies. It projected that both innovation & green tech imports will have detrimental impacts on CO2 emissions, suggesting that adopting green technology and keeping up with technical breakthroughs could assist in lessening the negative ecological effects of mineral mining. The first step in Table 2 is to look at the variables' stationarity to determine each independent variable's assumed values. The [39] unit root evaluation is employed for this. Next, the ARDL bounds test [40] is used to determine whether there is a longtime relationship among the variables, and [41] the co-integration test is used to confirm the results. The bootstrap phase of the ARDL test [42] is used to obtain the coefficients. The concept of error correction used for

**Table1. Data measurements.**

| The factor | Description | Sign | Origin | Predicted pattern of effects |
|---|---|---|---|---|
| Carbon emissions | After mining emissions | CO2 | China Financial Report and IEA along with other world databases | A reliant term |
| Legislative law | Legal actions effectiveness | Law | The World Greenhouse Initiative | + |
| Openness | Governance openness | Openness | The World Greenhouse Initiative | + |
| Green economic growth index | Comprehensive green growth | GEGI | Generated with the structure given by (Eraker &Shaliastovich, 2008) | + |
| Foreign direct investment | FDI in mining of minerals | FDI | The International Monetary | + |
| Technology innovation | Importance of technology | GTI | World Intellectual Property Organization | + |
| Green Patent | Eco-friendly mineral extracting technology imports | Green patent applications | The trade map | + |

**Table 2. Each stages estimating procedure.**

| Step | Estimation pattern | Statistical test method |
|---|---|---|
| 1 | Assess stability change | Unit root test |
| 2 | Extended correlation studies | ARDL bounds test |
| 3 | Collaborative evaluation | Test Co-integration test |
| 4 | Utility of variable prediction | ARDL technique |
| 5 | Evaluation of temporary parameters | Error correction term |
| 6 | Verify for models security | Diagnostic tests |
| 7 | The reliability rating | Changing dependent variable |

immediate correlation estimation. After these approximations, diagnostic procedures carried out to assess the sustainability of the experimental model, guaranteeing its dependability. The empirical results also tested for robustness. This broad framework ensures a complete & trustworthy correlation assessment of the variables under inquiry. This is shown in the Fig 2 upper chart, which shows the total amount of emissions from humans (dark blue line) and the proportional buildup of CO2 in the atmosphere (red line) since 1750.

This graphic illustrates how human activity has a cumulatively increasing effect on atmospheric CO2 levels. The percentage of these total emissions that remain in the atmosphere displayed in the lower chart, which adds further context. This emphasizes how, in spite of natural sinks' absorptive capacities, a substantial and ongoing parts of emissions contribute to climate change (Fig.3).

## 4. Results and discussion

### 4.1. Estimation results

This shows the remarks obtained from our empirical approximations. The first step is to verify the variables' normal distribution. The Zivot-Andrew's unit root test is shown in Table 3, which provides insight into the variables' stability properties.

The parameters show stationary behavior after the first differencing, pointing to a possible cointegration link between them, according to Table 3's findings. We use the bootstrap ARDL test to verify that this cointegration relationship exists. Furthermore, the Hatami-J's test applied to provide additional validity and robustness to the cointegration result. Tables 4 and 5 give the entire results of the mutual integration assessments, confirming the factors' long-term association.

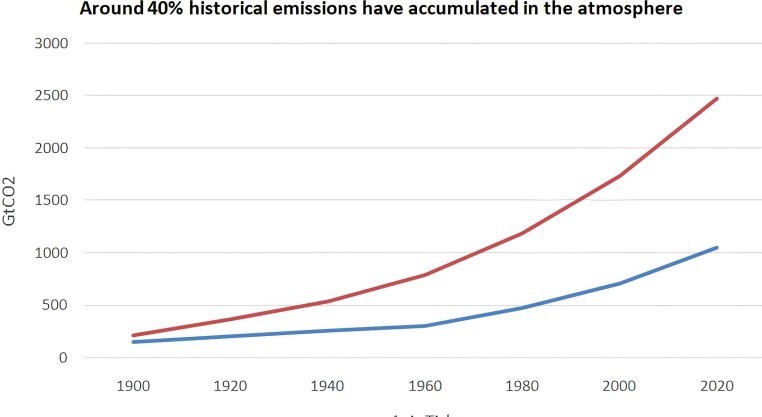

**Fig 2. Around 40% historical emissions have accumulated in the atmosphere near here.**

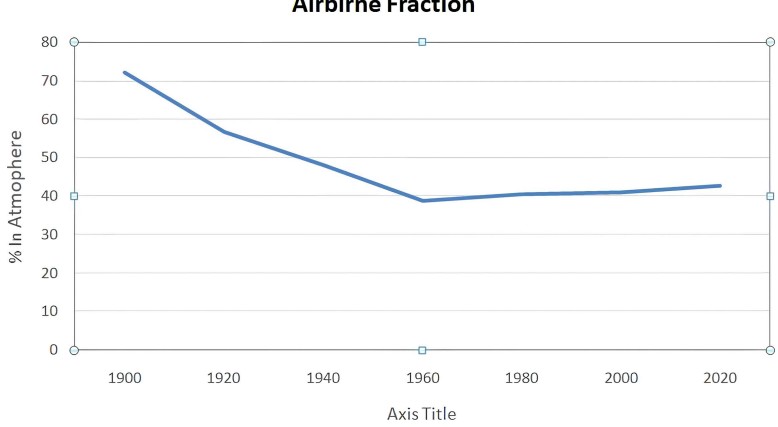

**Fig 3. Airborne Fraction near here.**

**Table 3. Zivot-Andrews unit root test.**

| Variable | I(1) | | I(0) | |
|---|---|---|---|---|
| | Value (Prob.) | Breakpoint | Value (Prob.) | Breakpoint |
| Carbon emissions | −0.943 (0.643) | 1994 | −5.488 (0.000) | 2003 |
| Legislative law | −1.463 (0.943) | 1996 | −7.196 (0.000) | 2000 |
| Openness | −1.865 (0.889) | 1995 | −6.890 (0.000) | 2003 |
| Green economic growth index | −2.326 (0.990) | 1997 | −10.065 (0.000) | 2001 |
| Foreign direct investment | −1.795 (0.779) | 1995 | −7.395 (0.000) | 2003 |
| Technology innovation | −2.493 (0.969) | 1993 | −9.044 (0.000) | 2005 |
| Green Patent | −1.633 (0.893) | 1998 | −8.500 (0.000) | 2007 |

Table 4 provides results of the Hatami-J test moving further with our analysis, we move on to the coefficient calculations using the ARDL method. Carefully displayed in Tables 5 and 6, correspondingly, are the outcomes of the ARDL method that incorporate both immediate and long-term factors. The quantitative model's linkages & fluctuations as seen over different periods in time laid forth in these spreadsheets.

More and more research on environmentally sustainable resource extracting has recently focused on the relationship between laws and regulations, the preservation of the environment, and economic development. As an example, [43] demonstrated how China's rigorous environmental laws have prompted innovations in clean mining technologies, which are in line with the country's aims to reduce carbon emissions. Regulation has a significant impact on technical advancement, as [44] showed through their demonstration of measurable emission reductions brought about by legislative changes that promoted the use of renewable energy in the mining industry [45]. In light of these results, we set out to investigate the mining industry's response to China's changing environmental regulations through a perspective of sustainable technology and economic behaviour.

The research indicates that the legislative law and openness have a positive impact on the long-term sustainability of China's mineral extracting industry. The mining extracting industry in China experiences a notable short-term reduction of about 0.41% and a sustained decrease of about 0.28% in carbon dioxide emissions in response to an increase of 1% in the legislative law variable. The mining extracting industry in China saw a substantial short-term drop of approximately 42% and a sustained decrease of approximately 21% in carbon dioxide emissions when openness increased by 1%. A strong legal framework and responsible leadership, which in turn promote efficient regulation, can explain the positive relationship. In contrast, the apparently paradoxical negative effects of both factors on emissions of CO2 in the possible the short and the long term could be due to the first modifications and obstacles that industries experience when they

**Table 4. ARDL bounded test-using Bootstrap.**

|  | Value | 1% | 5% | 10% |
|---|---|---|---|---|
| **Pesaran F** | 9.434** | 4.89 | 4.32 | 3.99 |
| **Dependent t** | −5.379** | 4.89 | 4.32 | 3.99 |
| **Independent F** | 8.014** | 6.79 | 5.39 | 3.69 |

**Table 5. Hatami-J test outcomes.**

|  | Probability | First Breakpoint | Second Breakpoint |
|---|---|---|---|
| **ADF** | −7.946 (1.0757) | 3999 | 3007 |
| **Zt** | −23.598 (1.049) | 3005 | 3009 |
| **ZA** | 278.314(0.036) | 3009 | 3009 |

**Table 6. ARDL results (short-term evidence).**

| Variables | Coefficient | Probability |
|---|---|---|
| Carbon emissions | −1.394 | 1.017 |
| Legislative law | −1.428 | 1.075 |
| Openness | −1.648 | 1.018 |
| Green economic growth index | 1.197 | 1.065 |
| Foreign direct investment | 1.099 | 1.029 |
| Technology innovation | −1.295 | 1.007 |
| Green Patent | −1.807 | 1.006 |

make efforts to adjust to more restrictive legal and voluntary methods. The industry's eventual alignment with rigorous rules and transparent methods may eventually overcome this short-term negative impact in terms of sustainability and preservation of the environment.

The estimation findings also show that a 1% rise to the green growth index has a substantial beneficial effect on lowering greenhouse gas emissions in China's minerals extracting sector. In particular, the results show that this kind of growth is associated with a significant drop in carbon dioxide emissions of about 0.66 percent in the next few years and 0.49 percent in the future. The green growth index's focus on promoting sustainable growth, social equality, and preservation of the environment is the likely explanation for the positive association. The green growth index demonstrates how policies that support green economic practices and environmentally friendly innovations encourage the mineral extracting industry to adopt improved and more sustainable techniques. Investing in eco-friendly innovations and procedures helps reduce carbon emissions, which is in accordance with worldwide as well as environmental objectives.

The research shows that FDI in China's mineral extracting sector does not have inherently green features, contrary to the beneficial effects seen with additional factors. According to the data, the mineral extracting sector's carbon dioxide emissions are going up with every 1% increase in FDI, with a short-term increase of over 21% and an extended rise of about 0.25%. Multiple variables contribute to the negative effect of FDI on carbon dioxide emissions. Globalisation may accelerate the widespread use of carbon-intensive techniques if foreign capital puts productivity and economic advancement ahead of rigorous environmental standards. More manufacturing and extracting could result from the flood of foreign capital, which would raise the carbon footprint even more. Additionally, it is interesting to see that innovation, as measured by patent applications, is actually increasing carbon dioxide emissions in China's mineral extracting sector. The results show that there are not enough green patents in the nation, since more patent applications lead to more carbon dioxide emissions. One possible explanation for this seemingly contradictory finding is that, instead of concentrating on sustainable development, many patents aim to improve productivity in extracting or manufacturing processes. There has to be a stronger push to encourage and reward ecologically responsible inventions in the mining sector due to the paucity of green patents that target environmentally friendly technology and procedures.

The results also prove that green technology imports helped China's mineral extracting sector cut its carbon footprint. Multiple variables contribute to the positive impact of adopting green technologies. According to the first point, these innovations usually follow the most up-to-date rules for protecting the environment and using energy efficiently. This helps lower the ecological impact of activities that extract minerals. Second, the spread of information and experience made possible by the import of cutting-edge, environmentally friendly technology from other nations allows businesses to develop more effective and less wasteful procedures. Finally, when in-house R&D is falling behind, imported green technologies can help speed up the move to eco-friendly business practices.

## 4.2 Diagnostics and robustness analysis

Systematic diagnostic methods used to evaluate the econometric strategy's sustainability. Table 7 provides a detailed presentation of the findings of these crucial tests, which confirm that the actual ARDL model is stable.

**Table 7. Results from Diagnostic assessment.**

| Tests | Statistics | Probability |
|---|---|---|
| Stability Ramsey RESET Test | 1.856 | 0.231 |
| ARCH Test | 1.776 | 0.496 |
| NormalityTest | 1.206 | 0.799 |
| BG Test serial correlation LM test | 0.433 | 0.191 |

Where BG stands for Breusch-Godfrey and ARCH for Autoregressive conditional heteroscedasticity

**Table 8. Robustness check.**

| Estimations | Factors | Coeff. | Prob. |
|---|---|---|---|
| Short-Term Evidence | Legislative law | −0.149 | 0.079 |
| | Openness | −0.295 | 0.043 |
| | Green growth index | −0.517 | 0.069 |
| Long-Term Evidence | Legislative law | −0.194 | 0.018 |
| | Openness | −0.315 | 0.055 |
| | Green growth index | −0.621 | 0.044 |

A diagnostic test must conduct to ensure the reliability of the results. To accomplish this, we estimated the coefficients while changing the dependent factor from carbon dioxide emissions throughout the mineral extracting sector to carbon dioxide throughout the metal sector in China. Table 8 summaries the findings of this diagnostic assessment, which only included the explanatory factors whose values were consistent with the results previously reported in Tables 6 and 7.

## 5. Conclusion and policy implications

### 5.1. Conclusion

The purpose of the present research is to measure the coefficients that affect China's mineral extracting-related CO2 emissions; these coefficients include lawfulness, openness, green economic growth, FDI, innovation, and green technology importation. The study, which used information from the China Statistical Yearbook and other worldwide databases covering the years 1989–2020, found that the legislative law and openness had a beneficial impact on the long-term viability of China's mineral extracting sector. Significant decreases in carbon dioxide emissions were associated with a 1% improvement in the legislative law and openness factors, demonstrating the results of effective regulation and accountability promoted by an open government and strong legal structures. Industry adaptation to extremely legal and transparent standards is expected to have a positive influence on sustainability in the long run, even though this is expected to have an initially paradoxical negative impact.

The study also demonstrated that when the green growth index increased by 1%, mineral extracting in China produced fewer carbon dioxide emissions. This research highlights the importance of policies that encourage environmentally conscious behaviours by fostering social equality, sustainable growth, and the preservation of the environment. The mineral extracting industry's carbon dioxide emissions increased due to FDI, which found to lack intrinsically green features. A possible explanation for this outcome is the rise in carbon-intensive processes generated by foreign investments that put a premium on economic development rather than rigorous environmental laws. In addition, the investigation of innovation uncovered an unexpected pattern: the mineral extracting industry in China experienced a surge in $CO_2$ emissions due to an uptick in patent applications. In light of this contradictory outcome, it is crucial to encourage environmentally conscious innovations. In contrast, the study confirmed that green technology imports help reduce carbon emissions, specifically mentioning how these technologies adhere to modern environmental requirements and how they help transfer knowledge.

### 5.2. Policy implications

Several policy recommendations can be included in considering the findings from this research to make China's mineral extracting sector more sustainable.

- Improving government openness and adherence to the legislative law must take prominence. In addition to encouraging environmental consciousness, an atmosphere that is advantageous to sustainable development can be created using strong legal structures and open practices. Policymakers should make strict rules that promote environmentally friendly procedures in the mineral extracting sector the primary goal. Also, governments should encourage and reward

environmentally friendly technology and practices since they are good for the economy and the environment. Investments in specific areas, new research and development programmes, and regulations that encourage these goals can all contribute to a more sustainable future.

- Lawmakers ought to consider ways to make sure that FDI is environmentally responsible so that it does not demage the environment. Offering incentives for environmentally friendly investments or strict environmental regulations for projects receiving funding from abroad are two possible ways to achieve this goal. It is necessary to promote innovations that prioritise sustainable development, especially in light of the unanticipated negative effect of innovation on carbon dioxide emissions. Policymakers must support green patents and work to create a regulatory environment that supports environmentally friendly innovations in the mining industry.

- Finally, lawmakers should encourage the import of cutting-edge, environmentally friendly technology from other nations to lower CO2 emissions. Establishing cooperative initiatives to transfer knowledge and expertise and implementing trade laws that encourage the import of environmentally friendly technologies are two ways to achieve this.

- To keep up with the ever-changing field of sustainability, researchers should use cutting-edge tools like big data analytics and artificial intelligence (AI) to investigate the mineral extraction industry's long-term viability. By analysing large datasets and identifying complex patterns in the environmental impact of the industry, artificial intelligence (AI) and big data can greatly improve the thoroughness and accuracy of assessments. Thanks to the integration of machine, learning algorithms into predictive modelling, researchers are now better able to predict future trends and potential mitigation strategies. In addition, to make the mineral extracting industry more sustainable, researchers should look into the effects of a carbon tax.

### 5.3. Limitations and future recommendations

Examining the interplay between laws and regulations, preservation of the environment and economic development in China's mineral extracting industry is an important area for future studies on the subject of law, sustainability, and the economy. Given China's continued prominence in the world's mineral supply, investigating how changing regulations can encourage sustainable practices while enhancing economic goals is crucial. Analysing the impact on local and global sustainability objectives, as well as the efficacy of recent legislative reforms to minimise environmental damage, could be one avenue for future research. Further research could examine the possibility of financial incentives for businesses to implement sustainability measures like environmentally friendly procurement and waste reduction. To promote sustainable mining procedures, research can also examine the significance of technology and innovation. This includes evaluating the way advanced techniques, like cleaner extracting technologies and digital evaluation, might reduce environmental effects while maintaining productivity.

In conclusion, future studies should examine how sustainable mining practices affect the communities that rely on them economically and how to ensure that everyone gets a fair share of the benefits. Focusing on these areas will help China achieve a more sustainable mineral resource sector by shedding light on the country's legal responsibilities, environmental concerns, and economic objectives.

### Author contributions

**Conceptualization:** Jing Wen.

**Data curation:** Dongxiao Xu, Jing Wen.

**Formal analysis:** Dongxiao Xu.

**Investigation:** Weijun Chen.

**Supervision:** Weijun Chen.

**Validation:** Yunzhe Chen.

**Visualization:** Yunzhe Chen.

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
