## [Decision Letter · Decision Letter 0]

15 Sep 2024

PONE-D-24-34068Assessing the Impact of Environmental Laws and Technological Advancements on Carbon Dioxide Mitigation in China's Mining IndustryPLOS ONE

Dear Dr. wen,

Thank you for submitting your manuscript to PLOS ONE. After careful consideration, we feel that it has merit but does not fully meet PLOS ONE’s publication criteria as it currently stands. Therefore, we invite you to submit a revised version of the manuscript that addresses the points raised during the review process.

We look forward to receiving your revised manuscript.

Kind regards,

Asif Khan, PhD Law

Academic Editor

PLOS ONE

Additional Editor Comments:

Reviewer 01

The title and abstract can be refined for conciseness and clarity, with the abstract requiring simplification to make the key findings more accessible. The introduction effectively sets the context but includes some repetition that could be eliminated, and the novelty of the study should be highlighted more clearly. The literature review is thorough but overly detailed, necessitating a more focused discussion on the most relevant studies and the gap this research addresses. In the methodology section, adding justification for the chosen statistical methods and a brief discussion on potential limitations would strengthen the study's credibility. The results and discussion are well-structured, though simplifying technical explanations and expanding on comparisons with similar studies could enhance reader understanding. the conclusion could be more concise, explicitly mention future research directions, and reinforce the study's contributions. the article writing style would benefit from simplification to improve readability, consistency in technical terminology.

Reviewer 02

Assessing the Impact of Environmental Laws and Technological Advancements on Carbon Dioxide Mitigation in China's Mining Industry

1. Clarity of Research Objectives: The article sets an important objective to evaluate the impact of environmental regulations and technological advancements on carbon mitigation in China's mining industry. However, the research objectives could be made more specific. It would be beneficial to clearly outline the key questions the study aims to answer and how these align with existing gaps in the literature.

2. Literature Review: While the article references several important works on environmental regulations and carbon mitigation, it would benefit from a more comprehensive review of recent developments in both international and Chinese mining sectors. For instance, it would be helpful to include a discussion on how international treaties, like the Paris Agreement, influence China's domestic policies and technological innovations in the mining sector.

3. Methodology: The article does not provide sufficient details regarding the methodology used to assess the impact of environmental laws and technology. A clear explanation of the data sources, selection criteria for technological advancements, and any empirical or qualitative methods employed would strengthen the overall rigor of the study.

4. Integration of Environmental and Technological Aspects: While the article discusses both environmental laws and technological advancements, the connection between the two remains somewhat disjointed. The study could be improved by explicitly linking how specific technological innovations in the mining industry are directly influenced by environmental policies and how these, in turn, contribute to carbon dioxide mitigation.

5. Case Studies and Examples: To enhance the article’s practical relevance, it would be useful to include more case studies or real-world examples from China's mining industry. For instance, specific technological interventions (e.g., carbon capture and storage technologies, or renewable energy integration in mining operations) and their measurable impacts on carbon mitigation should be discussed in greater depth.

6. Quantitative Analysis: The article could benefit from a more robust quantitative analysis. Incorporating statistics on carbon emissions reduction rates, technological adoption rates, and the effectiveness of environmental laws could provide a clearer picture of the mining industry's progress toward carbon mitigation.

7. Discussion and Policy Recommendations: While the article identifies important trends in China’s efforts to mitigate carbon emissions, the discussion would be stronger with a more in-depth analysis of the challenges and opportunities ahead. What policy recommendations can be made based on the findings? Are there any barriers to technological adoption in the mining industry that need addressing? These considerations would add depth to the conclusion.

8. Conclusion: The conclusion effectively summarizes the study, but it could be improved by more directly addressing the research question. It would also be helpful to outline potential future areas of research, particularly in the context of evolving technologies and tightening environmental regulations in China.

Overall, the article is a valuable contribution to understanding carbon mitigation in China’s mining industry but would benefit from more specific data, case studies, and a stronger linkage between the environmental and technological aspects discussed.

Reviewers' comments:

Reviewer's Responses to Questions

**Comments to the Author**

1. Is the manuscript technically sound, and do the data support the conclusions?

Reviewer #1: Yes

Reviewer #2: Yes

2. Has the statistical analysis been performed appropriately and rigorously? 

Reviewer #1: Yes

Reviewer #2: Yes

3. Have the authors made all data underlying the findings in their manuscript fully available?

Reviewer #1: Yes

Reviewer #2: Yes

4. Is the manuscript presented in an intelligible fashion and written in standard English?

Reviewer #1: Yes

Reviewer #2: Yes

5. Review Comments to the Author

Reviewer #1: The title and abstract can be refined for conciseness and clarity, with the abstract requiring simplification to make the key findings more accessible. The introduction effectively sets the context but includes some repetition that could be eliminated, and the novelty of the study should be highlighted more clearly. The literature review is thorough but overly detailed, necessitating a more focused discussion on the most relevant studies and the gap this research addresses. In the methodology section, adding justification for the chosen statistical methods and a brief discussion on potential limitations would strengthen the study's credibility. The results and discussion are well-structured, though simplifying technical explanations and expanding on comparisons with similar studies could enhance reader understanding. the conclusion could be more concise, explicitly mention future research directions, and reinforce the study's contributions. the article writing style would benefit from simplification to improve readability, consistency in technical terminology.

Reviewer #2: General Feedback:

The manuscript demonstrates thorough research and provides a comprehensive analysis of the topic. The argument is well-structured, and the use of evidence is strong in most sections. However, there are a few areas that require attention to improve clarity and enhance the reader’s understanding.

Strengths:

The introduction provides a solid foundation and clearly sets up the context for the discussion.

The argument is logically developed, and the use of data is convincing.

The language is formal and appropriate for the target audience.

Areas for Improvement:

Clarity in Argumentation: In certain sections, the argumentation could be refined for better clarity. Some points are repeated, and a more concise presentation would make the overall narrative stronger.

Depth of Analysis: While the evidence is well-chosen, certain areas lack depth. Consider expanding on key arguments to fully explore their implications.

Transitions Between Sections: The flow between sections feels abrupt at times. Smoother transitions could help guide the reader more effectively.

Conclusion: The conclusion is somewhat brief. It could benefit from a stronger summary of the key findings and a discussion of their broader implications.

Recommendations:

Rework certain sections for better clarity and conciseness.

Deepen the analysis in critical parts of the argument.

Ensure that transitions between sections are smooth to maintain a logical flow throughout the manuscript.

6. PLOS authors have the option to publish the peer review history of their article (what does this mean? ). If published, this will include your full peer review and any attached files.

**Do you want your identity to be public for this peer review?** For information about this choice, including consent withdrawal, please see our Privacy Policy .

Reviewer #1: No

Reviewer #2: No

---

## [Author Response · Author response to Decision Letter 0]

29 Nov 2024

Response

Reviewer # 1

1) We appreciate the comments. The title and abstract were revised to be more concise, clear, and easy to understand, with an emphasis on the main findings to make them more readable. To improve the introduction's ability to communicate the study's innovation and significance, we cut out unnecessary details and focused on its innovative contributions. These revisions improved the paper's flow and made it more effective.

2) We greatly appreciate your feedback. The revised literature review is more targeted, drawing attention to the most pertinent studies while making the research gap that this study addresses simple to understand (Literature last paragraph).

3) We appreciate your insightful feedback. In response, we expanded the methodology section to explain why the statistical methods used were suitable for addressing the research question and producing reliable results.

4) Discussion section increased and essential details are provided by comparing with similar studies. Robustness and diagnostic tests are applied (section 4.2) to make our results more accurate than previous studies.

5) Thanks for the helpful comments. The conclusion has been modified to be shorter; more explicit recommendations for future research were pointed out, and the study's contributions were emphasised. Further steps were taken to enhance readability by simplifying the language and ensuring that technical terminology was consistent throughout the paper. These changes clarified the text as a whole and responded to the concerns highlighted.

Reviewer #2

1) Thanks for your feedback. We revised the contribution section to address all issues. Literature section is also revised to understand how our work is different from previous studies. Study gap is provided in literature reviews last paragraph.

2) Thank you for your helpful comments. Consequently, we broaden the scope of our literature study to incorporate current events in both the global and Chinese mining industries, with an emphasis on the ways in which international accords, such as the Paris Agreement, impact China's internal policies. We discuss how these deals stimulate technical advancements in China's mining industry with the goal of decreasing carbon emissions and environmental damage.

3) We greatly appreciate your feedback. We updated the methodology section to better explain the process of evaluating the effects of environmental regulations and technological advancements in response. We defined the data sources and selected technological advancements (e.g., their impact on carbon mitigation, reliability, and significance).

4) Your feedback is greatly appreciated. Our response was to revise the work so that it more clearly linked environmental law to mining technology developments. By describing how these technologies directly contribute to carbon dioxide mitigation, we further demonstrated the causal relationship between policy-driven innovation and its environmental impact. With these modifications, we hope to give a more unified narrative and strengthen the study's main point.

5) Your insightful feedback is greatly appreciated. In response, we strengthened the article's practical significance by adding more case studies and real-world examples from China's mining industry in the literature and discussion section. Among the particular technical interventions we explored were the use of renewable energy sources in mining and carbon capture and storage (CCS) systems. A closer look was given to these instances, with an emphasis on the quantifiable effects they had on carbon mitigation.

6) Thanks for your suggestion. Discussion section increased and essential details are provided by comparing with similar studies. Robustness and diagnostic tests are applied (section 4.2) to make our results more accurate than previous studies.

7) We greatly appreciate your feedback. In response, we rewrote the article to give a more comprehensive overview of the possibilities and threats China faces in its fight against mining-related carbon emissions. In addition, we discussed the policy suggestions that emerged from our research, which emphasized the importance of encouraging sustainable mining practices and stronger enforcement of environmental laws.

8) We appreciate your comment. To resolve this issue, we revised the last section to address the research question more explicitly and highlight how our results assist in achieving a balance between regulations, environmental concerns, and economic objectives in China's mineral resource extraction.

---

## [Decision Letter · Decision Letter 1]

8 Jan 2025

Assessing the Impact of Environmental Laws and Technological Advancements on Carbon Dioxide Mitigation in China's Mining Industry

PONE-D-24-34068R1

Dear Dr. Chen,

We’re pleased to inform you that your manuscript has been judged scientifically suitable for publication and will be formally accepted for publication once it meets all outstanding technical requirements.

Kind regards,

Asif Khan, PhD Law

Academic Editor

PLOS ONE

Additional Editor Comments (optional):

Reviewers' comments:

Reviewer's Responses to Questions

**Comments to the Author**

1. If the authors have adequately addressed your comments raised in a previous round of review and you feel that this manuscript is now acceptable for publication, you may indicate that here to bypass the “Comments to the Author” section, enter your conflict of interest statement in the “Confidential to Editor” section, and submit your "Accept" recommendation.

Reviewer #1: All comments have been addressed

Reviewer #2: All comments have been addressed

2. Is the manuscript technically sound, and do the data support the conclusions?

Reviewer #1: Yes

Reviewer #2: Yes

3. Has the statistical analysis been performed appropriately and rigorously? 

Reviewer #1: Yes

Reviewer #2: Yes

4. Have the authors made all data underlying the findings in their manuscript fully available?

Reviewer #1: Yes

Reviewer #2: Yes

5. Is the manuscript presented in an intelligible fashion and written in standard English?

Reviewer #1: Yes

Reviewer #2: Yes

6. Review Comments to the Author

Reviewer #1: (No Response)

Reviewer #2: I am writing to confirm that the article titled "Assessing the Impact of Environmental Laws and Technological Advancements on Carbon Dioxide Mitigation in China's Mining Industry" has been thoroughly reviewed and is deemed suitable for publication.

7. PLOS authors have the option to publish the peer review history of their article (what does this mean? ). If published, this will include your full peer review and any attached files.

**Do you want your identity to be public for this peer review?** For information about this choice, including consent withdrawal, please see our Privacy Policy .

Reviewer #1: No

Reviewer #2: **Yes: ** Dr Syed Raza Shah Gilani

---

## [Editor Report · Acceptance letter]

PONE-D-24-34068R1

PLOS ONE

Dear Dr. Chen,

I'm pleased to inform you that your manuscript has been deemed suitable for publication in PLOS ONE. Congratulations! Your manuscript is now being handed over to our production team.

Kind regards,

on behalf of

Dr. Asif Khan

Academic Editor

PLOS ONE